# Optoelectronic Torque Measurement System Based on SAPSO-RBF Algorithm

**DOI:** 10.3390/s24051576

**Published:** 2024-02-29

**Authors:** Kun Xia, Yang Lou, Qingqing Yuan, Benjing Zhu, Ruikai Li, Yao Du

**Affiliations:** Department of Electrical Engineering, University of Shanghai for Science and Technology, Shanghai 200093, China; xiakun@usst.edu.cn (K.X.); yuanqq@usst.edu.cn (Q.Y.); 212221688@st.usst.edu.cn (B.Z.); 22221546@st.usst.edu.cn (R.L.); 223371787@st.usst.edu.cn (Y.D.)

**Keywords:** photoelectric torque measurement, quadrant photoelectric sensor, measurement of minute signals, torque calibration, SAPSO-RBF algorithm

## Abstract

The torque is a significant indicator reflecting the comprehensive operational characteristics of a power system. Thus, accurate torque measurement plays a pivotal role in ensuring the safety and stability of the system. However, conventional torque measurement systems predominantly rely on strain gauges adhered to the shaft, often leading to reduced accuracy, poor repeatability, and non-traceability due to the influence of strain gauge adhesion. To tackle the challenge, this paper introduces a photoelectric torque measurement system. Quadrants of photoelectric sensors are employed to capture minute deformations induced by torque on the rotational axis, converting them into measurable voltage. Subsequently, the system employs the radial basis function neural network optimized by simulated annealing combined with particle swarm algorithm (SAPSO-RBF) to establish a correlation between measured torque values and standard references, thereby calibrating the measured values. Experimental results affirm the system’s capability to accurately determine torque measurements and execute calibration, minimizing measurement errors to 0.92%.

## 1. Introduction

Torque is one of the fundamental parameters in rotational mechanical systems, playing a pivotal role in various engineering activities which involve the use of rotating machinery [1,2,3]. Its significance spans across industries such as mechanical manufacturing [4], automotive manufacturing [5], aerospace engineering [6], maritime sectors [7], and other related fields. Within shaft-driven systems, the torque value is intricately linked to mechanical power performance, energy consumption, lifespan, and safety measures [8,9]. Hence, the accurate, timely, and reliable measurement of torque holds significant importance in ensuring the overall safety and stability of mechanical devices and systems [1].

Currently, torque sensors are classified into 12 types based on different physical principles. These include resistive strain gauge types, click types, magneto-elastic types, distributed fiber grating types, surface acoustic wave types, magneto strictive types, magneto Backhausen noise types, magnetoelectric types, photo elastic types, photoelectric effects, laser Doppler-based types, and laser speckle torque sensors. Based on the principles of these sensors, researchers worldwide have investigated various measurement systems to achieve high-precision torque measurements. Among them, the most prevalent is the strain-based torque measurement system. In [10,11], extensive studies have been conducted on this method, employing Wheatstone bridge configurations to counteract axial and bending strains while compensating for temperature variations. Despite its simple installation process, this method exhibits limitations in measurement precision, susceptibility to external interferences, and relatively slow response speed. Apart from this method, in [12], the relationship between the deflection angle of the magnetization vector and torque is established based on the principle of electromagnetic effects and a non-contact torque measurement system utilizing Hall sensors is proposed. In [13], a torque measurement device based on the Hall effect is also introduced. It involves placing a pair of magnets on a rotating shaft, where torque-induced deformation causes relative movement between the magnets, altering the magnetic field intensity. The torque is measured by assessing the changes in the magnetic field intensity. In [14], improvements to the Surface Acoustic Wave (SAW)-based torque sensor are introduced, and a torque measurement system based on Surface Transverse Wave (STW) resonators is proposed. This enhancement is aimed at addressing drawbacks such as complex signal demodulation and limited operational range. And in [15], a novel non-contact vertical induction torque measurement system is proposed. The system consists of a stator and a rotor, with the stator comprising excitation coils and two receiving coils. When torque generates a torsional angle, different induced voltages occur between the two receiving coils. The variation in induced voltage is utilized to measure the rotor’s angular change, thereby quantifying the torque. While these measurement systems have undergone improvements, they still rely on electromagnetic principles, rendering them susceptible to environmental electromagnetic interference that can impact measurement precision.

Compared to the methods discussed earlier, torque measurement methods based on optical principles inherently offer advantages, allowing for higher precision and sensitivity [1]. For instance, in [16], a non-contact torque measurement system is designed based on zebra-tapes and optical sensors. This system involved connecting a pair of zebra-tapes with relative angular phases of zero to the end of a shaft. Additionally, two optical sensors were mounted on a non-rotating bracket. Torque was measured by quantifying the relative torsional angle of the shaft through the phase difference between the two pulse signals. In [17], a photoelectric torque sensor is designed based on the principle of phase difference to measure the torque of a ship’s shaft. Two photoelectric discs are installed on the shaft, each equipped with two photoelectric sensors. When the shaft rotates and the light-blocking gear in the photoelectric disc obstructs the laser, the sensor outputs a high signal; conversely, when the light-blocking gear does not obstruct the laser and it illuminates the photodiode, the sensor outputs a low signal. The phase difference between the high and low signal levels is used to measure the torque. Subsequently, a sliding window algorithm is employed to reduce measurement errors and improve accuracy. In [18], a high-sensitivity, compact torque sensor based on Fiber Bragg Grating (FBG) technology is also presented. This sensor comprises a torque-sensitive bending structure and two diagonally arranged optical fibers, incorporating an embedded fiber Bragg grating sensor. It can achieve high resolution and good linearity characteristics. And in [19], a novel non-contact sensor based on the principle of Optical Coherence Displacement Measurement (OCDM) is proposed for measuring micro-torque on a rotating shaft. Optical sensors are employed to measure the linear displacement of the shaft surface relative to the measuring arm, obtaining the optical coherence displacement. Subsequently, a unique mapping relationship between OCDM and torque is established. This relationship involves designing a laser engraving system to process scales on the shaft surface, representing the torsional angle. This approach can resist the influence of negative torque radial vibrations, achieving a high-precision torque measurement. However, these methods require specialized circuit design and complex computational formulas to derive the final torque value, making the process intricate and cumbersome.

In addition to enhancing measurement devices, researchers have proposed various calibration algorithms aimed at optimizing measurement outcomes and minimizing errors. For instance, in [20], a Deep Neural Network (DNN) algorithm was utilized to calibrate sensors’ multi-axis output signals, effectively reducing errors. Meanwhile, in [21], a Fusion Decoupling Algorithm based on Least Squares Support Vector Machine Regression (LSSVR) was introduced to refine the measurement values derived from sensors. Although these algorithms can significantly enhance measurement outcomes, their downside lies in their complexity and relatively slow computational speed.

In this paper, the system addresses torque measurement through advancements in both torque sensors and calibration algorithms, presenting a novel photoelectric torque measurement system. Concerning the sensor technology, this system integrates quadrant photoelectric sensors as the primary measuring devices. These sensors, constituted by an array of photodiodes, offer distinct advantages including fundamental simplicity, high-precision, and rapid response characteristics. Positioned on the rotating shaft, these sensors feature LED light sources that centrally illuminate the four quadrants, resulting in equivalent current outputs from each quadrant in a static state [22]. Once the shaft undergoes torque-induced deformation, causing a corresponding shift in the light spot vertically, the currents from the four quadrants alter accordingly. Subsequently, the processing circuit converts these varying currents into voltage values for output. Concerning the calibration algorithm, this system employs a Particle Swarm Optimization (PSO) algorithm, incorporating simulated annealing (SA), to optimize a Radial Basis Function (RBF) neural network, forming the SAPSO-RBF algorithm. Compared with deep learning networks such as CNN [23], RBF networks, characterized by their uncomplicated structure, robust approximation capabilities, and computational efficiency, find widespread applications in fuzzy recognition and nonlinear fitting [24]. Within this paper, the RBF neural network is utilized for the calibration of torque measurement values. Simultaneously, the SAPSO algorithm fine-tunes the parameters of the RBF network. This innovative approach addresses the RBF network’s dependency on parameters, thereby enhancing the algorithm’s optimization efficacy.

This paper is structured as follows: Section 2 introduces the measurement principle of the system as well as the algorithm principle, while Section 3 explains the overall architecture of the system including the hardware and software design. Section 4 focuses on the construction of the whole calibration algorithm model and process, and Section 5 verifies the system experimentally, derives the measurement results and compares the measurement errors under different conditions. Lastly, Section 6 provides the conclusions of the entire system.

## 2. The Principle of the System

### 2.1. The Principle of Measurement

The measurement system employs four-quadrant photoelectric sensors to convert torque-induced deformations into voltage signals, requiring sensor modules characterized by high sensitivity and precision. Specifically, the chosen sensors are TE Connectivity’s QP50-6-42u SD2 products (TE Connectivity, Shanghai, China). These sensors feature a transimpedance amplifier circuit capable of converting current to voltage, facilitating differential signal $V_{x}$ (obtained by subtracting the bottom signal from the top) and $V_{y}$ (calculated by subtracting the left signal from the right). Moreover, the QP50-6-42u SD2 also generates a voltage signal, $V_{z}$, representing the cumulative output of all four-quadrant diode signals.

The sensor modules and LED light source are affixed onto four measuring arms, uniformly and symmetrically positioned along the circumference of the shaft to eliminate the bending effect of the shaft. Under static conditions, the LED light source emits light, forming a centralized light spot within the four quadrants. When torque impacts the shaft, it induces tangential deformation, causing displacement of both the LED light source and its corresponding spot. This displacement generates current signals from the four quadrants, which, subsequently, undergo processing by the processing circuit. These current signals are transformed into three distinct voltage outputs, namely $V_{x}$, $V_{y}$, and $V_{z}$, and then relayed to the signal acquisition board. The schematic illustrating the installation of LED light source and photoelectric sensors on the measuring arms is shown in Figure 1. 

The three voltage signals’ values can be calculated in the following formulas:(1)$$V_{x}=\left(V_{1}+V_{4}\right)-\left(V_{2}+V_{3}\right)$$
(2)$$V_{y}=\left(V_{1}+V_{2}\right)-\left(V_{3}+V_{4}\right)$$
(3)$$V_{z}=V_{1}+V_{2}+V_{3}+V_{4}$$
where $V_{x}$ represents the disparity between the voltage signals produced by the right and left quadrants of the photodiode; $V_{y}$ denotes the difference between the signals from the top and bottom quadrants of the photodiode; $V_{z}$ signifies the cumulative signal generated by all four quadrants of the photodiode; $V_{1}$ to $V_{4}$ represent the voltage values obtained by processing the current from each quadrant through the circuit. Measurement of the aforementioned output voltages facilitates the deduction of the relative movement of light spot across the four-quadrant sensor. This deduction is achieved through the application of the following formulas:(4)$$X=\frac{\left(V_{1}+V_{4}\right)-\left(V_{2}+V_{3}\right)}{V_{1}+V_{2}+V_{3}+V_{4}}=\frac{V_{x}}{V_{z}}$$
(5)$$Y=\frac{\left(V_{1}+V_{2}\right)-\left(V_{3}+V_{4}\right)}{V_{1}+V_{2}+V_{3}+V_{4}}=\frac{V_{y}}{V_{z}}$$
where Y represents the relative displacement along the vertical axis, essential for deriving the applied torque. Additionally, it captures the relative displacement in the horizontal direction, denoted as X, used for subsequent analysis of the coupling effect of the thrust on the torque applied to the axis. The calculation of torque is expressed as:(6)$$T=\frac{G\cdot Y\cdot I_{P}}{r}$$
where G represents the shear modulus of the shaft material; r stands for the radius of the shaft section; $I_{P}$ denotes the polar moment of inertia, a crucial physical quantity solely contingent upon the shape and dimensions of the cross-section. $I_{P}$ plays a pivotal role as one of the essential coefficients in computing the resistance against torsion, and is expressed by:(7)$$I_{P}=\int_{A}r^{2}dA=\frac{\pi d^{4}}{32}\approx 0.1d^{4}$$
where d represents the diameter of the shaft section.

### 2.2. SAPSO-RBF Calibration Algorithm

#### 2.2.1. RBF Neural Network

The RBF (Radial Basis Function) neural network, characterized by a straightforward three-layer architecture, possesses the capability to approximate any continuous function with arbitrary precision. Within this structure, the intermediate layer functions as the hidden layer, employing the RBF function as its activation mechanism. Defined as a monotonically increasing function of the Euclidean distance in space between a given point x and a specific center c, the RBF function embodies various forms, with the Gaussian function being the most prevalent [24]. This function’s representation is shown as follows:(8)$$p_{j}=\frac{1}{\sqrt{2\pi }}\exp\left(-\frac{{\left(x-c_{j}\right)}^{2}}{2{\sigma_{j}}^{2}}\right)\;j=1,\;2,\;\ldots ,\;n$$
where x represents the input sample; $c_{j}$ denotes the center of the *j*-th node in the hidden layer; $\sigma_{j}$ stands for the width of the *j*-th node in the hidden layer.

While the standard RBF neural network model boasts a simple structure, fast computation, and robust nonlinear processing capabilities, its efficacy heavily relies on determining essential parameters during the learning phase. Consequently, identifying the optimal parameters for the network becomes paramount. This paper employs the Simulated Annealing Particle Swarm Optimization (SAPSO) algorithm to fine-tune and optimize the network parameters.

#### 2.2.2. SAPSO Optimization Algorithm

The Particle Swarm Optimization (PSO) algorithm originated from observations of bird foraging behaviors and has since found widespread applications in parameter optimization and other areas. Within the framework of the PSO algorithm, each particle’s position represents a potential solution in the computational process. Following each iteration, the particles within the swarm progressively converge towards individual historical best solutions, culminating in the pursuit of the overall optimal solution for the collective [25]. The iterative nature of the PSO algorithm involves continual adjustments to the particles’ positions and velocities, updated as follows: (9)$$v_{id}^{k+1}=\omega \times v_{id}^{k}+c_{1}\times r_{1}\times \left(p_{id}^{k}-x_{id}^{k}\right)+c_{2}\times r_{2}\times \left(g_{d}^{k}-x_{id}^{k}\right)$$
(10)$$x_{id}^{k+1}=x_{id}^{k}+v_{id}^{k+1}$$

The PSO algorithm is an evolutionary algorithm based on a global search strategy, renowned for its fast convergence. However, its drawback emerges when exploring solution spaces, especially in high-dimensional spaces, where the initialization of particles significantly affects the optimization outcome. Sometimes, particles exhibit oscillations near the optimal solution, which may be a local optimum. This occurrence impacts the accuracy of the algorithm [26].

In addressing this issue, this paper introduces the Simulated Annealing (SA) algorithm and combines it with the PSO algorithm to form the SAPSO algorithm. The Simulated Annealing algorithm possesses the capability to accept poorer solutions with a certain probability, enabling it to escape local optima. Under conditions where the initial temperature is adequately high and the temperature decreases slowly enough, it can converge to the global optimum with a probability of 100% [27].

The Simulated Annealing algorithm employs the Metropolis criterion. At each step of the algorithm, a perturbation generates a new candidate solution, S2, randomly. If this new solution causes a decrease in the objective function from the current solution, S1, it is acceptable. However, if the new solution leads to an increase in the objective function, its acceptance is determined by an exponential probability. The probability of accepting the new solution, denoted by P, is defined as:(11)$$P=\left\{ \begin{array}{cc} \exp\left(-\Delta f/T\right) & \Delta f>0\\ 1 & \Delta f\leq 0 \end{array}\right.$$
where T represents the current temperature value; ∆f stands for the increment in the objective function between the new solution and the current solution. If ∆f≤0, then S2 is accepted as the new current solution. Otherwise, the acceptance probability P for S2 is calculated. Subsequently, a uniformly distributed random number, r, is generated in the (0, 1) interval. If P>r, S2 is accepted as the new current solution; otherwise, the current solution S1 is retained [27]. Upon accepting the new solution, the annealing operation as shown below is executed to decrease the temperature value:(12)T(t+1)=βT(t)
where β represents the annealing factor, which ranges between (0, 1).

After integrating the simulated annealing algorithm, the SAPSO algorithm not only exhibits strong global search capabilities, but also incorporates the probability of accepting inferior solutions when generating new ones. This ensures population diversity, aiding in escaping local optima and enhancing both convergence speed and optimization precision.

## 3. System Architecture

### 3.1. System Hardware Design Overview

The optoelectronic torque measurement system proposed in this paper can be broadly divided into two main components: the rotor and the stator. The rotor section comprises four sets of quadrant photoelectric sensors, measurement arms with their fixed apparatus, precise LED light sources, wireless power reception coils, a power supply board, two signal acquisition boards, and a signal transmission board. All modules are mounted on the shaft. The stator section consists of a wireless power transmission coil, a signal reception board, and a host computer. The rotor section employs four four-quadrant photoelectric sensors, strategically positioned along the shaft to detect torque-induced deformation. This deformation is then converted into voltage signals, subsequently routed to the signal acquisition board. Comprising two distinct boards, each managing signals from two sensors, the acquisition board meticulously samples and processes the signals. The processed signals are then transmitted through RS232 to the signal transmission board. Following this, the transmission board relays the signals via Wi-Fi to the signal reception board located at the stator end. The signal reception board seamlessly communicates with the upper computer using the RS485 physical bus and the Modbus-RTU communication protocol. The entire system is powered by the wireless power transmission coil at the stator end. Upon induction of current by the wireless reception coil at the rotor end, it is rectified and stabilized by the power supply board before being distributed individually to each module at the rotor end. A schematic diagram depicting the overall hardware design is shown in Figure 2.

### 3.2. Software Design

The software components primarily encompass sampling programs for the two signal acquisition boards in the rotor section, the wireless communication protocol for the signal transmission board, and the validation program for the signal reception board in the stator section, along with the host computer program.

Figure 3 shows the software workflow of the entire measurement system. Upon the initialization of all processors, corresponding peripherals such as ADCs, Wi-Fi modules, etc., are initialized first. Following this initialization phase, the signal transmission board initiates a synchronized signal transmission to both signal acquisition boards at one-second intervals. This action enables the simultaneous commencement of sampling operations on the two signal acquisition boards. Once the two boards receive the synchronization signal, they activate timers to conduct AD sampling at the specified sampling frequency. Simultaneously during sampling, a sliding average filter is applied to the data to mitigate the impact of external random errors. After the 1 s sampling period elapses, each signal acquisition board simultaneously sends the sampled data to the signal transmission board in sequential order. To ensure data integrity, a validation byte is appended to each data stream. Upon receiving the data, the transmission board verifies the correctness of each acquisition board’s data by checking these validation bytes. If no errors are detected, the transmission board arranges all channels of sensor data in sequence, combines them with a frame header, forming a data frame. This data frame is transmitted via Wi-Fi to the signal reception board at the stator end. Upon receiving the data, the signal reception board first verifies the frame header for accuracy. Once confirmed, it extracts voltage data from each sensor channel, converts it back into torque measurement values, and then sends these torque values to the host computer via the Modbus protocol and the RS485 bus for data processing and display.

## 4. Construction of the SAPSO-RBF Calibration Algorithm Model

Due to inherent errors in the torque values acquired by the system, it becomes necessary to construct an SAPSO-RBF calibration algorithm model to rectify these measurements. Within the model, the parameter settings of the Radial Basis Function (RBF) neural network play a critical role in its overall performance. The determination of the network’s hidden layer center vectors ($c_{j}$) and standardization constants ($\sigma_{j}$) poses a complex and challenging problem in this context. Therefore, this paper employs the SAPSO algorithm to optimize the aforementioned parameters. The overall process is shown in Figure 4, with specific steps outlined as follows:To initialize the positions and velocities of each particle within the population, assuming an RBF neural network with InNum input layer nodes, HiddenNum hidden layer nodes, and OutNum output layer nodes, the particle’s dimensionality is defined as d = HiddenNum. This means that all center vectors of the hidden layer nodes are regarded as values for the particles;Initialize the temperature for the Simulated Annealing algorithm. For each particle, assign an initial temperature T1;Utilize Mean Squared Error (MSE) as a metric to assess the performance of the RBF network, represented by the formula as follows;
(13)$$MSE=\frac{1}{k}\sum_{j=1}^{k}e^{2}\left(j\right)=\frac{1}{k}\sum_{j=1}^{k}{\left(t_{j}-y_{j}\right)}^{2}$$
where $t_{j}$ represents the expected output; $y_{j}$ stands for the actual output.Since each particle’s position represents the parameters of the RBF network, Formula (13) is utilized as the fitness function for particles. The fitness value for each particle is computed by substituting the particle’s position into the RBF network. Initially, the initial position of each particle serves as its initial individual best position (pbest0), and the position of the particle with the minimum fitness among all particles is designated as the initial global best position (gbest0);Using the algorithm for iteration, during each iteration, the fitness value for each particle is calculated. If this value is less than the fitness value associated with the particle’s previous pbest, the particle’s position is updated to become its new pbest. If the fitness value is less than the fitness value associated with gbest, the particle’s position is updated to become the new gbest;As the gbest calculated in the previous step could potentially be a local optimum, accordingly, based on the principles of the simulated annealing algorithm, add a random disturbance to each particle and calculate the minimum fitness value for the perturbed particle, as well as the overall best position based on the perturbed values. Continuing, use the following formula to calculate the fitness value increment before and after;
(14)$$\Delta E=f^{t+1}(x_{i})-f^{t}(x_{i})$$
where $f^{t+1}\left(x_{I}\right)$ represents the minimum fitness value after perturbation. $f^{t}\left(I\right)$ stands for the fitness value of the gbset before perturbation;If Δ*E* ≤ 0, the new solution is accepted as the current solution, and the temperature value is updated according to Formula (12). Otherwise, with the probability described in Formula (11), the fitness value is accepted. If accepted, the temperature is updated as Formula (12); otherwise, no temperature update occurs;Check for the termination condition. If not met, update the particles according to Formulas (9) and (10) and repeat the process from step 4 to step 6. If the condition is met, terminate the process and apply the parameters of the gbest to the network;Finally, apply the RBF neural network optimized by the algorithm for torque calibration.

## 5. Torque Calibration Experiment

### 5.1. Data Collection and Processing

In order to validate the measurement performance of this system, an experimental setup as shown in Figure 5 was constructed.

Before utilizing the calibration algorithm for data optimization, it is essential to train the network parameters following the procedures outlined in Section 4. Therefore, the initial step involves collecting a training dataset for torque data. Within this dataset, the inputs are the measured torque values, while the corresponding outputs are the standardized torque values. To derive these standardized values, the T40 torque flange sensor from HBM company, acclaimed for its 0.03% linear accuracy, is utilized. Employing shear stress technology, this sensor measures the shear force applied to an object, thereby accurately measuring torque stress. In this experiment, the torque load spans from 0 N·m to 200 N·m, with increments of 10 N·m. At each measurement point, 20 sets of data are collected, resulting in a total sample size of 420. Due to varying magnitudes in the collected torque data, normalization of the samples becomes crucial to prevent computational saturation. To achieve this, the following formula is used to confine the data within [0, 1] range:(15)$$X_{out}=\frac{X-X_{\min}}{X_{\max}-X_{\min}}$$
where *X* represents the measured torque values within the training dataset; $X_{\min}$ denotes the minimum value within the sample set; $X_{\max}$ signifies the maximum value present in the sample set.

### 5.2. Calibration Results

After processing the data, it was input into the calibration algorithm described in Section Four for training. Additionally, for comparison purposes, SAPSO-RBF and PSO-RBF algorithms were trained using the same set of hyperparameters. The specific hyperparameters for the algorithms are detailed in Table 1. And the RBF network utilized the same inputs, outputs, and number of hidden layers as listed in the table. The change in fitness values of global extrema for the first two algorithms during training over iterations is shown in Figure 6.

In Figure 6, it can be observed that initially, the fitness value of the SAPSO-RBF algorithm is higher than that of the PSO-RBF. This occurs because the simulated annealing algorithm probabilistically accepts suboptimal solutions during each iteration to prevent the algorithm from becoming stuck in local optima, resulting in a slower decline in fitness value. With the increase in iteration count, the PSO-RBF algorithm converges around 230 iterations, while the SAPSO-RBF algorithm continues to search for the optimal solution until approximately 310 iterations before entering convergence, ultimately achieving a fitness value superior to that of the PSO-RBF algorithm.

To verify the calibration effects of the algorithms, the three calibration algorithms mentioned above are applied to calibrate the measured torque values. We selected five sets of torque measurement values for validation, namely 0, 40, 80, 120, 160, and 200 Nm, with each set consisting of 20 data points. The calibration results for torque are shown in Figure 7.

In Figure 7, it can be observed that after processing with the SAPSO-RBF algorithm, the fluctuation in torque values for each set is significantly smaller compared to the other two algorithms. This indicates that the SAPSO-RBF algorithm exhibits a much better calibration effect, notably enhancing measurement accuracy. Figure 8 shows a more intuitive comparison of the calibration performance of the three algorithms by showing the absolute value of error between each data test point and the standard torque values. Table 2 provides a summary of the root mean square error (RMSE) for each group of torque values in Figure 8, along with the maximum error among the 20 measurement points in each group.

The summary from Table 2 indicates that, after optimization with the SAPSO-RBF algorithm, the maximum error and RMSE for each group of torque values are the smallest. This suggests that, compared to the other two algorithms, the SAPSO-RBF algorithm exhibits the optimal calibration performance.

After verifying the calibration effects of the algorithm, the data measurement range is expanded. Torque measurement points are selected at intervals of 10 Nm within the range of 0–200 Nm, with each group consisting of 20 measurement points. The absolute values of relative errors for all samples (except for samples with a value of 0) under the three scenarios are compared, as shown in Figure 9. And Table 3 summarizes the comparison of the MAPE and the time consumption for algorithm training and running.

Table 3 shows that the calibration of the PSO-RBF algorithm reduces the MAPE by 1.247% compared to the traditional RBF network, while the SAPSO-RBF algorithm achieves a more significant reduction of 2.687%. This indicates that the proposed SAPSO-RBF algorithm for optimizing the RBF network has successfully achieved the intended optimization effect, resulting in a decreased measurement error. Regarding training time, the traditional RBF network, which uses the pseudoinverse method for parameter solving, does not require training. However, this leads to poor calibration performance and higher errors. Although the training time for the SAPSO-RBF algorithm is slightly longer than that of the PSO-RBF algorithm, it results in a 1.44% reduction in calibration error. Moreover, the three algorithms have similar running times, falling within an acceptable range.

## 6. Conclusions

In this paper, an optoelectronic torque measurement system that employs highly precise and sensitive four-quadrant photoelectric sensors to measure the torque applied to the shaft is introduced. The entire system comprises rotor and stator modules, facilitating data transmission via Wi-Fi. The rotor module is responsible for collecting sensor outputs, packaging them into data frames, and transmitting them via Wi-Fi. Conversely, the stator module receives the data from the rotor module and sends it to the host computer for monitoring. This setup accomplishes a separation between sampling and monitoring and enables non-contact torque measurement. Additionally, the entire system employs the SAPSO-RBF algorithm to calibrate torque measurement values, thereby further enhancing measurement accuracy. Experimental validation confirms the system’s ability to accurately obtain torque values and reduce errors through algorithmic calibration, providing a novel solution for non-contact torque measurement. However, our system still has some gaps compared to the state-of-the-art torque sensors currently available. Additionally, the presence of both stator and rotor components makes the installation and maintenance of the entire system more complex.

In future work, we intend to take the following measures to improve our study. Firstly, we will optimize the hardware and mechanical structure, improving the structure and installation of the measuring arm to enhance its static stability. Additionally, we aim to integrate the rotor’s circuitry onto a single circuit board, reducing the complexity of installation and maintenance. Secondly, acknowledging the prevailing challenge of temperature influence on measurement accuracy, we intend to incorporate a temperature sensor to monitor ambient temperature in real-time. To counteract the potential impact of temperature variations, a dedicated temperature compensation algorithm will be implemented. Lastly, we will continue researching more advanced torque calibration algorithms to further reduce measurement errors.

## Figures and Tables

**Figure 1 sensors-24-01576-f001:**
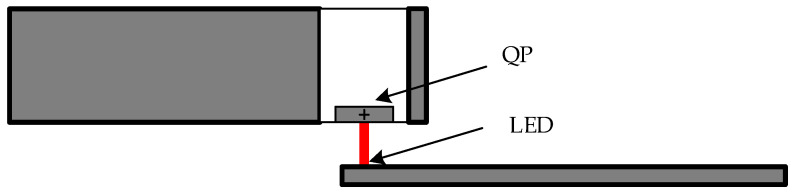
LED light source and photoelectric sensor installation diagram.

**Figure 2 sensors-24-01576-f002:**
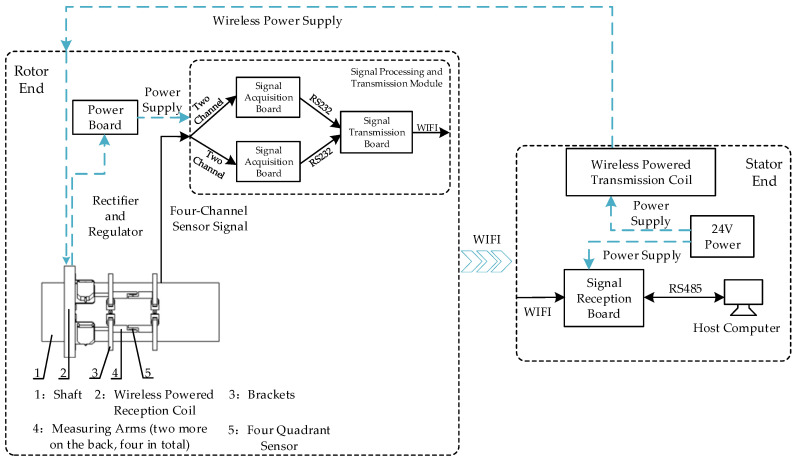
Schematic diagram of the overall hardware design of the photoelectric torque measurement system.

**Figure 3 sensors-24-01576-f003:**
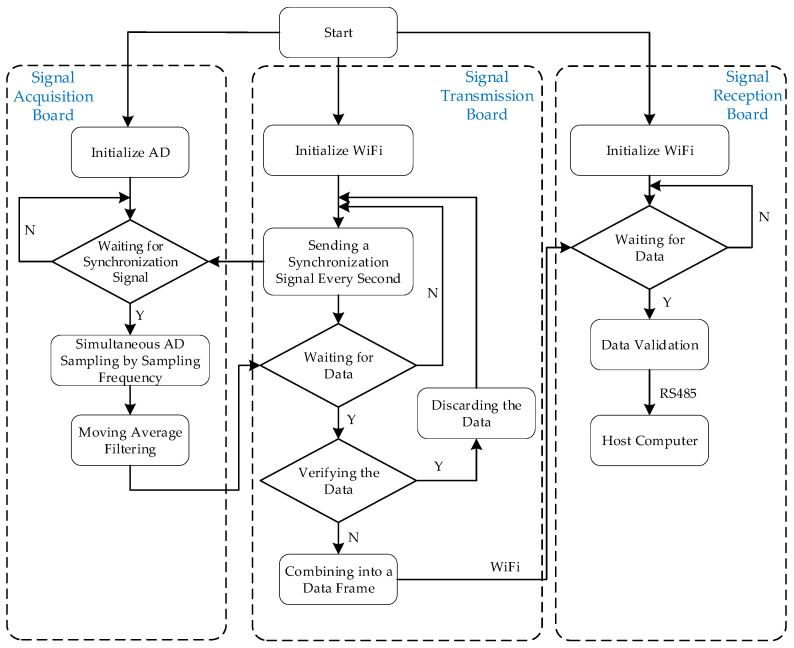
Software workflow of the measurement system.

**Figure 4 sensors-24-01576-f004:**
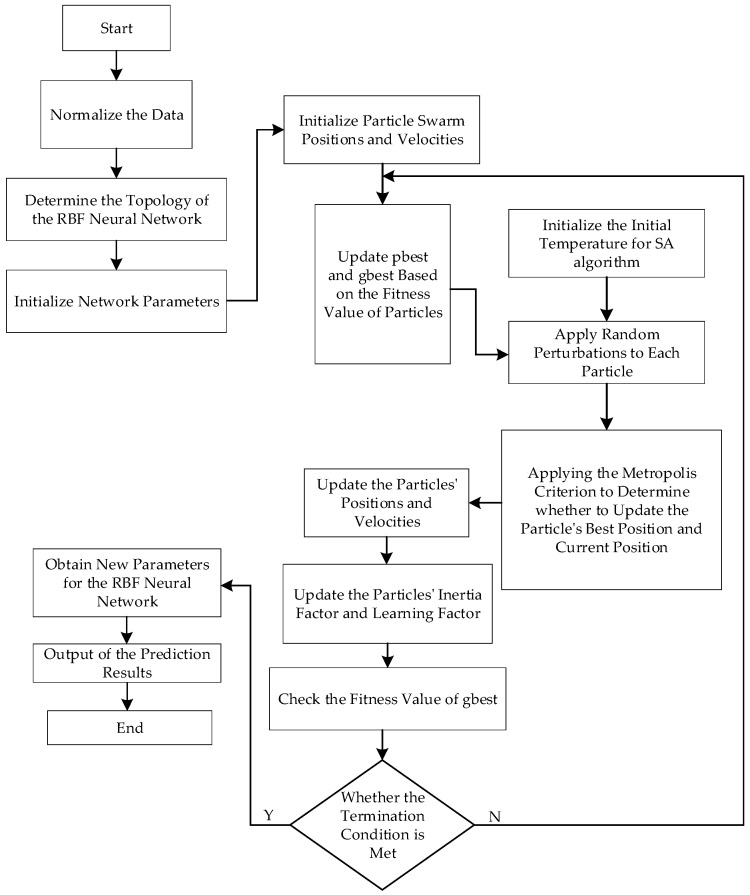
Flowchart of the SAPSO-RBF algorithm.

**Figure 5 sensors-24-01576-f005:**
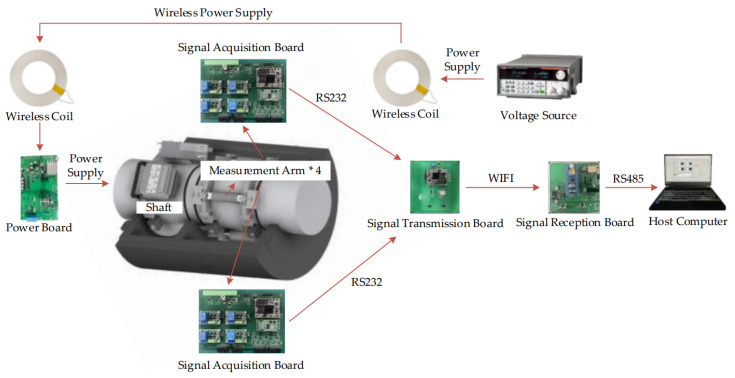
Experimental configuration for validating the measurement performance of the system.

**Figure 6 sensors-24-01576-f006:**
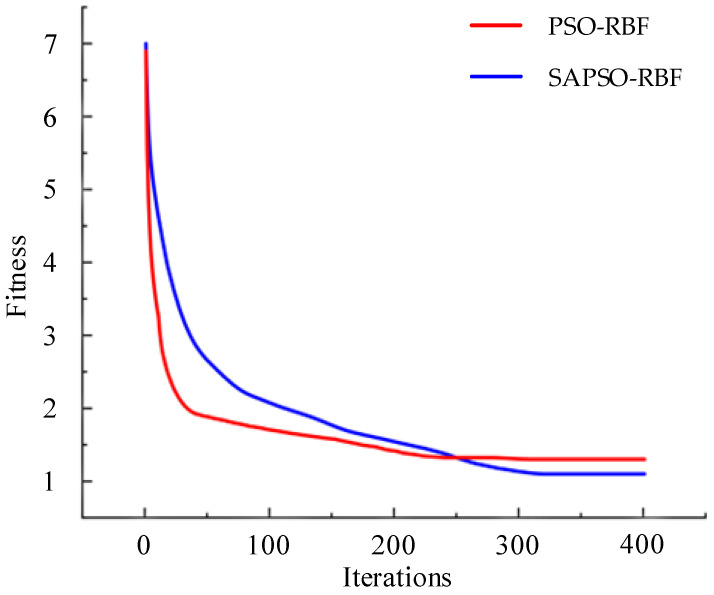
Fitness curves of the PSO-RBF and SAPSO-RBF algorithms in the training process.

**Figure 7 sensors-24-01576-f007:**
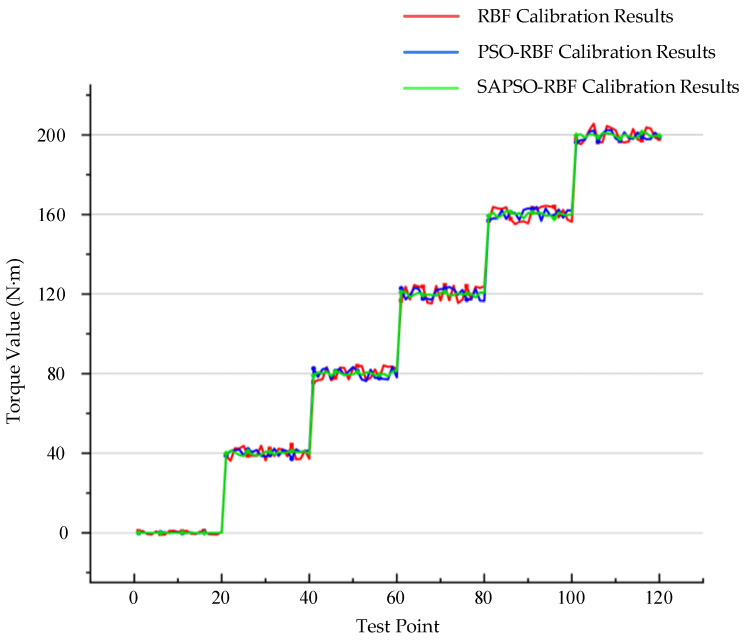
Torque calibration results for the three algorithms.

**Figure 8 sensors-24-01576-f008:**
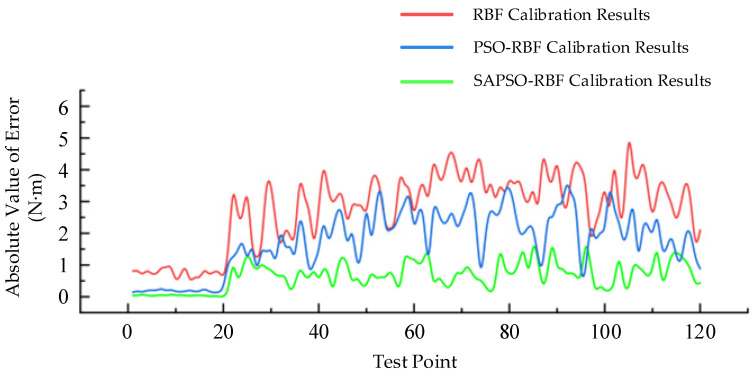
Absolute value of error between calibrated torque measurement values and standard values.

**Figure 9 sensors-24-01576-f009:**
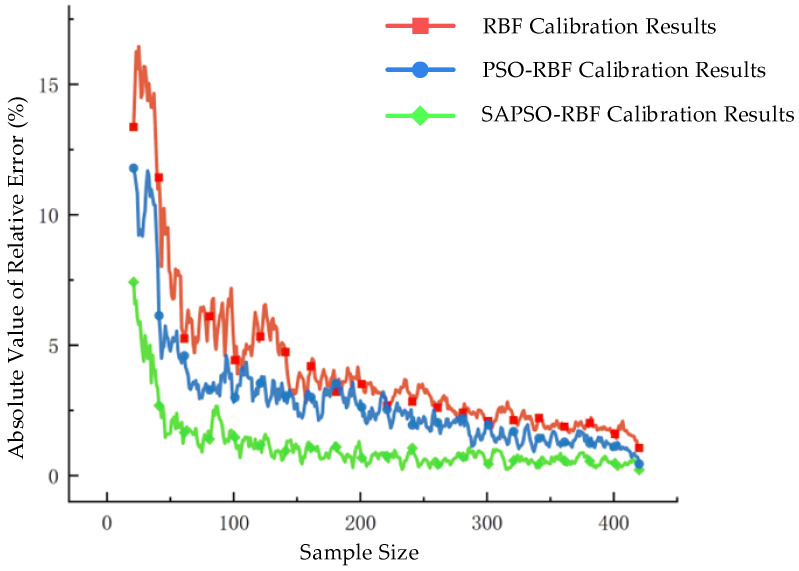
Comparison of error results under the three scenarios.

**Table 1 sensors-24-01576-t001:** Algorithm training hyperparameters.

Hyperparameter	Value
Input Layer Number	1
Hidden Layer Number	15
Output Layer Number	1
Particle Swarm Dimension (D)	15
Particle Number (N)	30
Weight (ω)	(0.4, 2)
Learning factor ($c_{1}$)	2.0
Learning factor ($c_{2}$)	2.0
Iteration Number (M)	400

**Table 2 sensors-24-01576-t002:** The maximum error and RMSE of the torque values for each test group.

Calibration Algorithm	Test Group											
	0		40		80		120		160		200	
RBF	1.03 ^1^	0.768 ^2^	3.824	2.61	4.432	3.145	4.682	3.782	4.867	3.476	5.179	3.359
PSO-RBF	0.263	0.189	2.77	1.53	3.685	2.426	3.557	2.626	3.754	2.338	3.753	2.117
SAPSO-RBF	0.076	0.042	1.42	0.81	1.423	0.836	1.726	0.824	1.942	1.057	1.802	0.953

^1^ For each test group, the first data represents the maximum error among the 20 measurement points in that group. ^2^ The second data in the pair represents the RMSE of the torque values in that group.

**Table 3 sensors-24-01576-t003:** Comparison of the three calibration algorithms for all samples.

Calibration Algorithm	Training Time (s)	Running Time (s)	MAPE (%)
RBF	—	0.0081	3.611
PSO-RBF	857.1	0.0076	2.364
SAPSO-RBF	863.4	0.0076	0.924

## Data Availability

Data are contained within the article.

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
