# Peer review of "Optoelectronic Torque Measurement System Based on SAPSO-RBF Algorithm"

_sensors, 2024, doi:10.3390/s24051576_

Round 1

Reviewer 1 Report

Comments and Suggestions for Authors

This paper introduces a photoelectric torque measurement system in which quadrants of photoelectric sensors are employed to capture minute deformations induced by torque on the rotational axis, converting them into measurable voltage. Subsequently, the system employs the SAPSO-RBF calibration algorithm to establish a correlation between measured torque values and standard references, thereby calibrating the measured values. Experimental results affirm the system's capability to accurately determine torque measurements and execute calibration, minimizing measurement errors to 0.92%.

The problem statement is clear, paper is written well but I feel the volume of the work is very low. The authors are advised to revise the paper significantly well and below are my concerns:

1. The Calibration Algorithms - RBF, PSO-RBF and SAPSO-RBF are compared w.r.to the respective parameters. Its natural that these algorithms perform better one over the other and the errors are minimized. Highlight the novelty or the factor which significantly impacts the error minimization.

2. As per section 4, include a section 4.1 to show the fitness value calculation using steps 4 to 6 and one final calculation that leads to determine pbest and gbest.

3. Why the impact of Algorithm training hyperparameters w.r.to torque value is not determined? I mean what will happen to torque value if Hidden Layer Number or Output Layer Number or Particle Swarm Dimension changes for SAPSO-RBF itself?

4. Why there is no optimization done within SAPSO-RBF for values as mentioned in comment 3?

5. The entire system comprises rotor and stator modules, facilitating data transmission via WiFi - Have you included the error model in WiFi environment?

6. In terms of simulation study, the results may be convincing but in real time, is the validity of the achieved results is correct or not?

7. If no for above comment, what will be the deviation % - if possible highlight it please.

8. Overall, work is good but need to be revised well as per the above said comments.

Comments on the Quality of English Language

Minor editing of English language required

Reviewer 2 Report

Comments and Suggestions for Authors

1. The main question addressed by the research is how to improve the accurate torque measurement.

2. The topic is relevant to the field of electrical engineering. This paper summarizes various problems of torque measurement in applications. It introduces a photoelectric torque measurement system.

3. Compared with other published materials authors employed photoelectric sensors to capture minute deformations induced by torque on the axis. They use SAPSO-RBF calibration algorithm.

4. Are there any limits of your system?

Did you compare economic aspects with conventional torque measurement systems?

Did you try your methodology also in industry?

5. The conclusions are consistent with the evidence presented and they address the main question.

6. References are appropriate.

Reference the papers and not the authors throughout the text. For example: In [1] it was proposed...

Include the digital object identifier (DOI) for all references where available.

Reviewer 3 Report

Comments and Suggestions for Authors

The authors have to incorporate the following suggestions for improving the readability of their work

1. What type of motor is considered and what is the rating of the Motor and load conditions  for  collecting the training data.

 2. What are the different conditions chosen for torque measurement  and what are the inputs given to the RBFN network for getting torque output.

 3. Samples of the input and output data used for training the RBFN network  are to be included for better clarity of the readers.

4. the error still seems to be a little high and can be reduced further considering more number of nodes in hidden layers.

Reviewer 4 Report

Comments and Suggestions for Authors

The article's theme is worthy of investigation.

1. Avoid using acronyms in the Title.

2. Kindly elaborate on the acronym SAPSO-RBF in the abstract and main body while using it the first time.

3. The introduction part should be more elaborated by integrating a rich text concerning the most recent advancements in the domain.

4. A Tabular comparative analysis of the present work with the recent state-of-the-art (On current interest) will be an add-on to the work.

5. Based on the Tabular analysis kindly highlight the Research gap followed by the novel initiatives to address the gap.

6. The Result part is the prime concern and not acceptable in the current state. The image quality of Fig (7-9) is not acceptable.

7. Kindly incorporate the relevant test cases supported by result graphics and relevant numeric analysis. In its current form, this hardly justifies the work.

Comments on the Quality of English Language

Moderate improvement required

Round 2

Reviewer 1 Report

Comments and Suggestions for Authors

The authors have carried out all suitable corrections suggested by the reviewers and they have improved the paper well. Hence the paper shall be accepted in present form.

Comments on the Quality of English Language

Minor editing of English language required

Reviewer 4 Report

Comments and Suggestions for Authors

The authors have marginally addressed the Reviewer's comments, The Revised manuscript now touches the journal standards.

Comments on the Quality of English Language

Moderate editing of English language required